

# Geomorphic regulation of floodplain soil organic carbon concentration in watersheds of the Rocky and Cascade Mountains, USA

Daniel N. Scott[1], Ellen E. Wohl[1]

[1]Department of Geosciences, Colorado State University, Fort Collins, CO, 80521, United States

*Correspondence to*: Daniel N. Scott (dan.scott@colostate.edu)

**Abstract.** Mountain rivers have shown the potential for high organic carbon (OC) storage in terms of retaining OC-rich soil. We characterize valley bottom morphology, floodplain soil, and vegetation in two disparate mountain river basins: the Middle Fork Snoqualmie, in the Cascade Mountains, and the Big Sandy, in the Wind River Range of the Rocky Mountains. We use this dataset to examine variability in OC concentration between these basins as well as within them, at multiple spatial scales. We find that although there are some differences between basins, much of the variability in OC concentration is due to local factors, such as soil moisture and valley bottom geometry. From this, we conclude that local factors likely play a dominant role in regulating OC concentration in valley bottoms, and that inter-basin trends in climate or vegetation characteristics may not translate directly to trends in OC storage. We also use analysis of OC concentration and soil texture by depth to infer that OC is input to floodplain soils mainly by decaying vegetation, not overbank deposition of fine, OC-bearing sediment. Geomorphology and hydrology play strong roles in determining the spatial distribution of soil OC in mountain river corridors.

## 1. Introduction

Terrestrial carbon storage plays an important role in regulating the global carbon cycle and the distribution of carbon between oceans, the atmosphere, long-term ($10^5 – 10^9$ years) storage in rock, and short- to moderate-term storage in the biosphere ($10^1 – 10^4$ years, including vegetation and soil) (Aufdenkampe et al., 2011; Battin et al., 2009). Soils, in particular, are a large organic carbon (OC) reservoir with significant spatial variability (Jobbágy et al., 2000; Schmidt et al., 2011), making them difficult to characterize in the context of global carbon cycling. It is essential to quantify the spatial variability of OC stored in the biosphere to constrain the effects of climate change on feedbacks between biospheric and atmospheric carbon storage (Ballantyne et al., 2012). To provide a more complete understanding of how the biospheric carbon pool may change in the future and guide management of soil OC, we seek to provide a better constraint on where carbon is stored in the biosphere and the processes that regulate that storage.

We focus here on river corridors, defined as channels, fluvial deposits, riparian zones, and floodplains (Harvey and Gooseff, 2015), which process, concentrate, transport, and store carbon (Wohl et al., 2017b). In the context of the carbon cycle, floodplains can act as a major component of the biospheric carbon pool (Aufdenkampe et al., 2011; Battin et al., 2009). Floodplain soil can act as a substantial pool of OC, indicating that floodplains may be disproportionately important compared to uplands in terms of carbon storage (D'Elia et al., 2017; Hanberry et al., 2015; Sutfin et al., 2016; Sutfin and Wohl, 2017; Wohl et al., 2012, 2017a). Mountainous regions, due to their high



primary productivity (Schimel and Braswell, 2005; Sun et al., 2004), may play a substantial role in the freshwater processing and storage of OC where they retain sediment and water along the river network (Wohl et al., 2017b). Even laterally constrained floodplains in mountainous drainages can store significant quantities of OC that can be mobilized during floods (Rathburn et al., 2017). It is important to understand the spatial distribution of OC to predict

its fate during floods and inform management to increase floodplain OC storage (Bullinger-Weber et al., 2014).

Floodplain OC enters river corridor soils via litterfall from vegetation and erosion of OC-bearing bedrock (Hilton et al., 2011; Leithold et al., 2016; Sutfin et al., 2016). OC inputs are either allochthonous, from upstream deposition of soil, particulate, and dissolved OC, or autochthonous, from riparian vegetation (Omengo et al., 2016; Ricker et al., 2013; Sutfin et al., 2016). As such, OC input can be regulated by vegetation dynamics and resulting

litter input, hydrologic and sediment transport regimes, and water chemistry.

OC concentration in soil is also regulated by the ability of carbon to sorb to soil particles and the ability of microbes to oxidize soil OC, which can be controlled by rhizosphere dynamics, moisture, and temperature. Sorption of OC to soil particles reduces OC lability and is controlled by grain size and resulting available surface area as well as the availability of calcium, iron, and aluminum (Kaiser and Guggenberger, 2000; Rasmussen et al., 2018).

Microbial processing oxidizes OC and represents the primary pathway by which soil OC returns to the atmosphere. In general, low temperatures and frequent saturation inhibit microbial activity and promote OC storage (Falloon et al., 2011; Jobbágy et al., 2000; Sutfin et al., 2016).

At inter-basin scales, hydroclimatic regime controls vegetation dynamics, moisture, and temperature, such that soil OC concentration in disparate regions can be approximately characterized by these predictors

(Aufdenkampe et al., 2011; Schimel and Braswell, 2005). However, at the scale of a single watershed, hydrology, ecology, and geomorphology play strong roles in determining soil texture, moisture, and microbial dynamics, in turn controlling OC storage in valley bottoms (Scott and Wohl, 2017; Sutfin and Wohl, 2017; Wohl and Pfeiffer, *In Press*). As such, a multi-scale approach must be taken to understanding spatial variation in OC concentrations in valley bottoms.

Here, we quantify spatial variations in OC concentration within two disparate river networks. This allows us to examine inter-basin hydroclimatic variation and intra-basin geomorphic and vegetation variation to understand the multi-scale controls on OC concentration.

### 1.1 Objectives and Hypotheses


Across a basin, it is uncertain whether OC concentration follows predictable longitudinal variation, or is controlled by local factors. Similarly, in a vertical floodplain soil profile, it is uncertain whether OC concentration follows a trend similar to uplands, with declining OC concentration with depth, or exhibits vertical heterogeneity as a result of OC-rich layers deposited by floods. It is also unclear whether OC in floodplain soils is dominantly

autochthonous or allochthonous. Floodplain soil OC source may be evident from the vertical heterogeneity of OC concentration, whereby dominantly autochthonous OC profiles should decline with depth whereas dominantly allochthonous OC profiles should exhibit vertical heterogeneity, reflecting episodic deposition. Our primary





objective here is to understand spatial variations in OC concentration both with depth in a soil profile and across a basin. By quantifying these variations, we hope to infer the processes that regulate OC deposition in floodplain soil.

By examining two disparate mountain river basins, we can quantify both inter-basin variation in OC storage as well as variation within each basin. We hypothesize that at an inter-basin scale, hydroclimatic regime and

resulting rate of litterfall inputs in the riparian zone (Benfield, 1997) will dominantly regulate OC concentration (H1). We define hydroclimatic regime as the combination of precipitation and temperature dynamics that result in the vegetation characteristics of a basin. At an intra-basin scale, we expect that valley bottom geometry and river lateral mobility will regulate floodplain sediment characteristics and vegetation dynamics. Thus, we hypothesize that soil OC concentration does not vary along predictable, longitudinal trends within mountain river basins, instead

being more dominantly controlled by local fluvial processes and valley bottom form (H2a). We hypothesize that geomorphic process and form determine soil texture and moisture, which in turn set the boundary conditions that regulate the sorption of OC to mineral grains (promoting stabilization) and the potential of OC to be respired by microbes (H2b). In terms of OC inputs to floodplain soils, we hypothesize that the source of OC is dominated by autochthonous vegetation and litter inputs in these basins (H3). As such, we expect OC to dominantly decline with

depth, only rarely exhibiting vertical heterogeneity that would represent allochthonous deposition from flooding.

## 2. Methods

This work was done alongside work presented in Scott and Wohl (*In Review*), and hence shares field sites, study design, GIS, and sampling techniques.

### 2.1 Field Sites

We quantified soil organic carbon concentrations to a depth of approximately one meter in the Big Sandy basin in the Wind River Range of Wyoming and the Middle Fork Snoqualmie basin in the central Cascade

Mountains of Washington (Figure 1). These basins represent distinct bioclimatic and geomorphologic regions, ranging from the wet, glacially influenced Cascades to the semi-arid Middle Rockies.

The MF Snoqualmie has a mean annual precipitation of 3.04 m (Oregon State University, 2004), 2079 m of relief over a 407 km$^2$ drainage area, and a mean basin slope of 60%. Topography in the MF Snoqualmie is largely glaciogenic, with wide, unconfined valleys at both high and low elevations. Streams range from steep, debris flow

dominated headwater channels to lower gradient, wide, laterally unconfined channels in its lower reaches. The lower reaches of the MF Snoqualmie have been clearcut extensively in lower reaches since in the early 1900s, although there is little logging activity today. Vegetation follows an elevation gradient. The talus, active glaciers, and alpine tundra at the highest elevations transition to subalpine forests dominated by mountain hemlock (*Tsuga mertensiana*) (above approximately 1500 m), but also including Pacific silver fir (*Abies amabalis*) and noble fir (*Abies procera*) in

the lower subalpine and montane zones (above approximately 900 m). Below the montane zone, uplands and terraces are covered by Douglas fir (*Pseudotsuga menziesii*) and western hemlock (*Tsuga heterophylla*), whereas active riparian zones are dominated by red alder (*Alnus rubra*) and bigleaf maple (*Acer macrophyllum*).



The Big Sandy is considerably drier than the MF Snoqualmie, but also exhibits broad, glacially carved valleys, especially in headwater reaches. It has a mean annual precipitation of 0.72 m (Oregon State University, 2004), 1630 m of relief over a 114 km$^2$ drainage area, and a mean basin slope of 25%. Herbaceous alpine tundra dominates higher elevations (above approximately 3100 m), while the subalpine zone (approximately 2900 to 3100

m) is characterized by forests of whitebark pine (*Pinus albicaulis*), Engelmann spruce (*Picea engelmannii*), and subalpine fir (*Abies lasiocarpa*). The montane zone (approximately 2600 m to 2900 m) is comprised dominantly of lodgepole pine (*Pinus contorta*). Only a small portion of this basin (approximately 1%) resides below 2500 m, where shrub steppe begins to dominate (Fall, 1994). Parklands and meadows are abundant in this basin, creating a patchy forest structure. Comparing this basin to the MF Snoqualmie provides bioclimatic contrast that allows us to

examine how floodplain soil OC concentrations vary across a range of stream morphologies and floodplain morphologic types in regions with differing precipitation, forest characteristics, and basin morphology.

**2.2 Study Design and Sampling**

We sampled the Big Sandy in summer 2016 and the MF Snoqualmie in summer 2017. During each sampling campaign, no large floods occurred and we observed no floodplain erosion or deposition. Across both basins, we cored a total of 128 floodplain sites to determine soil OC concentration. Cores were collected as a series of individual soil samples at both regular and irregular depth increments.

**2.2.1 Big Sandy**

The sparse vegetation in the Big Sandy basin enabled us to use a combination of a 10 m DEM and satellite imagery to manually map the extent of the valley bottom along the entire stream network and delineate valley bottoms based on confinement. We defined unconfined valley bottoms as those in which channel width occupied no

more than half the valley bottom, and confined valley bottoms as those in which channel width occupied greater than half the valley bottom. Within each confinement stratum, we stratified the stream network by five drainage area classes to produce a total of ten strata, ensuring even sampling across the basin. Within each of the resulting ten strata, we randomly selected 5 reaches, producing a total of 50 sample sites throughout the basin. Due to access issues, we sampled 48 out of the 50 randomly located sites. We supplemented these with 4 subjectively located sites

that we felt enhanced our ability to capture variation throughout the drainage based on observations in the field, resulting in a total of 52 sampled sites.

**2.2.2 Middle Fork Snoqualmie**

The MF Snoqualmie River basin is larger than the Big Sandy and has extensive, low-gradient floodplains in its downstream reaches. These extensive floodplains display high spatial variability in vegetation, surface water, grain size, and estimated surface age, based on aerial imagery and ground reconnaissance. To ensure an unbiased characterization of these heterogeneous floodplains, we used aerial imagery, a 10 m DEM, and pictures from field





reconnaissance to delineate the floodplain into patch categories: fill channels (abandoned channels that have had enough sediment deposited to prevent an oxbow lake from forming), point bars (actively accreting surfaces on the inside of bends), wetlands (areas with standing water in imagery that are not obviously oxbows), oxbow lakes (abandoned channels dammed at the upstream and downstream ends to form a lake), and general floodplain surfaces

(surfaces that cannot be classified into any of the above categories). Within each of these five categories, we randomly selected six points at which to take soil cores.

We also stratified the entire MF Snoqualmie stream network by channel slope into four strata. Within each channel slope strata (hereafter referred to as simply slope strata), we randomly selected ten reaches to collect a single floodplain soil core, resulting in 40 randomly located sample sites.

To supplement randomly sampled sites and accommodate for the infeasibility of accessing two of the randomly sampled sites along the stream network, we also subjectively selected sample sites in places that we felt enhanced the degree to which our sampling captured the variability present among streams in the basin. This resulted in a total of 46 sites stratified by slope, 38 of which were randomly sampled, in addition to 30 sites stratified by floodplain type.

## 2.3 Reach-Scale Field Measurements

At each sampled reach (100 m or 10 channel-widths, whichever was shorter), we measured channel geometry and other characteristics, although our measurements were not consistent across all basins because field

protocol evolved during the course of the study. In both basins, we measured confinement, valley bottom width, and channel bed slope. We additionally measured bankfull width and depth in the MF Snoqualmie. We did not measure channel characteristics for sites stratified by floodplain type in the MF Snoqualmie, since they did not correspond to a single reach of channel, as did sites stratified by slope in much more confined valleys.

In the MF Snoqualmie, we also categorized channels by planform and dominant bedform (Montgomery and

Buffington, 1997). We defined planforms as either: straight, where the channel was generally confined and significant lateral migration was not evident, meandering, where lateral migration was evident but only a single channel existed, anastomosing, where vegetated islands separated multiple channels, and anabranching, where a single dominant channel existed with relict channels separated by vegetated islands. We further classified channels as being either multithread (anastomosing or anabranching) or single thread (straight or meandering). Because

logging records are inconsistent and likely inaccurate in the MF Snoqualmie (based on the frequent observation of past logging activity where none was recorded in Forest Service records), we noted whether signs of logging, such as cut stumps, cable, decommissioned roads or railroads, or other logging-associated tools were found near the reach.

We chose a representative location on the floodplain for each sampled site, based on visual examination of

vegetation type, soil surface texture, surface water presence, and elevation relative to the bankfull channel elevation (floodplain sites stratified by type in the MF Snoqualmie were sampled as close to the randomly sampled point as possible). Once a location was chosen, we extracted a 32 mm diameter soil core using an open-sided corer (JMC



Large Diameter Sampling Tube). Due to our adaptive methodology, we sampled soil OC slightly differently in the Big Sandy basin versus the MF Snoqualmie. In the Big Sandy, we cored in irregular increments, generally 25-30 cm. After analyzing data from the Big Sandy basin, we realized that sampling in regular increments would make analysis more versatile. As a result, we switched to extracting soil samples at regular, 20 cm increments in the MF

Snoqualmie Basin. Cores were taken to refusal (i.e., coarse gravel or other obstructions preventing further soil collection) or a depth of approximately 1 m. Five cores in the Big Sandy, 12 cores in the MF Snoqualmie sites stratified by slope, and 11 cores in the MF Snoqualmie sites stratified by floodplain type did not reach refusal. When no sand or finer sediment was present in the valley bottom (only occurred in headwater channels of the MF Snoqualmie), we recorded negligible OC concentration. Once soil samples were removed from the ground, they

were placed in ziplock bags and frozen within 72 hours (most samples were frozen within 8 hours) and kept frozen until analysis.

**2.4 Measuring Soil OC and Texture**

To measure the concentration of organic carbon in soil samples, we used loss-on-ignition (LOI). We first defrosted samples for 24-48 hours at room temperature. Once defrosted, we thoroughly mixed samples to ensure the most homogenous sample possible. We then subsampled 10-85 g of soil from each sample for analysis. Using crucibles in a muffle furnace, we dried samples in batches of 30 for 24 hours at 105°C to determine moisture content and remove all non-structurally held water. Following the guidelines suggested by Hoogsteen et al. (2015), we then

burned samples for 3 hours at 550°C to remove organic matter. By comparing the weight of the burned samples with that of the dried samples, we obtained an LOI weight.

After performing LOI, we used burned samples to perform texture-by-feel to determine the USDA soil texture class and estimated clay content (Thien, 1979). To convert LOI weight to OC concentration, we used the structural water loss correction of Hoogsteen et al. (2015) using clay content estimated from soil texture. This

correction considers water held by clay that may not evaporate during drying, but will evaporate during burning. It also estimates the proportion of the LOI weight that is OC.

One potential confounding factor in LOI is carbonates that may burn off during ignition, adding to the LOI weight while not being organic matter. In lithologies where carbonates are rare (e.g., granitoid rocks like those found in the upper MF Snoqualmie and entire Big Sandy basins), this is a relatively negligible issue. However, some of our

soil samples came from parts of the MF Snoqualmie basin draining rocks of the western mélange belt, including argillite, graywacke, and marble. We tested samples for the presence of carbonates to determine whether our LOI methods would be sufficient to accurately determine OC concentration. We randomly chose 10 soil samples of a total of 110 that drained rocks that could include carbonates and submitted them to the Colorado State University soil testing laboratory for CHN furnace analysis (Sparks, 1996), which yielded data on the proportion of carbonates

by mass in those samples. On average, those samples contained calcium carbonate concentrations of 0.97% (95% confidence interval between 0.96% and 0.97%), and the percentage of the total carbon in those samples comprised of inorganic carbon was, on average, 8.6% (95% confidence interval between 8.51% and 8.78%). From this, we





concluded that the amount of carbonate in the samples draining potentially carbonate-bearing rocks was low enough that LOI was likely to still be accurate. Consequently, we analyzed all soil samples using LOI to obtain OC concentration.

## 2.5 GIS and Derivative Measurements

After fieldwork in each basin, we collected the following data for each reach using a GIS platform: elevation, drainage area, land cover classification and canopy cover from the National Land Cover Database (Homer et al., 2015), and the mean slope of the basin draining to each reach (including hillslopes and channels). Utilizing
drainage area at each reach and field-measured channel gradient, we calculated an estimated stream power as the product of drainage area, channel gradient, and basin-averaged precipitation. We utilized a 10 m DEM for all GIS topographic measurements. To estimate clay content for each sample, we used median values for assigned USDA texture classes. To obtain estimated clay content, moisture, and OC for each core, we calculated an average weighted by the percentage of core taken up by each soil sample. For samples stratified by floodplain in the MF
Snoqualmie, we categorized floodplain types into those with standing water (wetlands and oxbow lakes) and those with no standing water (all other types).

## 2.6 Statistical Analyses

All statistical analyses were performed using the R statistical computing software (R Core Team, 2017). We conducted all analyses on three modeling groups, based on the variables measured in each group. In the MF Snoqualmie, we grouped observations by stratification type, separating observations stratified by channel slope from observations stratified by floodplain type. We separated these two groups from all observations in the Big Sandy, which were measured consistently. We modeled OC concentration and soil texture with a mixed effects linear
regression using individual soil samples (i.e., the individual samples that make up a core) as sample units (n = 103 for MF Snoqualmie stratified by slope, 89 for MF Snoqualmie stratified by floodplain type, and 101 for Big Sandy). We modeled the sampled site as a random effect, acknowledging that individual soil samples within a single core are likely non-independent. We use profiled 95% confidence intervals on effect estimates ($\beta$) for fixed effects to evaluate variable importance in mixed effects models.

To gain further insight at the reach-scale, we also modeled average OC concentration and soil moisture at each measured site using multiple linear regression. We modeled soil moisture at the reach scale because we felt that our single snapshot of moisture conditions was better represented as a site-level average. We first performed univariate analysis between each hypothesized predictor and the response, utilizing mainly comparative Wilcoxon rank-sum tests (Wilcoxon, 1945) or correlational Spearman correlation coefficient statistics. We utilize a Holm
multiple-comparison correction (Holm, 1979) for pairwise comparisons. During this filtering, we also view boxplots or scatterplots as appropriate to discern which variables appear to have anything other than a completely random relationship with the response. We then utilize all subsets multiple linear regression using the corrected Akaike Information Criterion as a model selection criteria (Wagenmakers and Farrell, 2004). We iteratively transformed





response variables to ensure homoscedasticity of error terms. To select a single best model, we utilized both Akaike weight based importance as well as parsimony to select a final, reduced model. We considered sample sizes, p values, and effect magnitudes in determining variable importance.

We also analyzed each core to determine whether there were buried, high OC concentration layers at depth.

We compared each buried soil sample to the sample above it using the criterion that a peak in OC at depth should have an OC concentration 1.5 times that of the overlying sample and be above 0.5% (Appling et al., 2014).

## 3. Results

Model results are presented in Table 1. Comparisons between basins and summaries of OC concentration,

moisture, and estimated clay content are shown in Figure 2.

### 3.1 OC Concentration

Of cores with more than a single sample, we found that 32% (7/22) of cores stratified by slope in the MF

Snoqualmie, 32% (8/25) of cores stratified by floodplain type in the MF Snoqualmie, and 6% (2/31) of cores in the Big Sandy exhibited OC concentration peaks at depth. Whether a soil sample was classified as an OC peak had no relation to estimated clay content in sites stratified by floodplain type ($p = 0.28$) or those stratified by slope ($p = 0.89$) in the MF Snoqualmie. In the Big Sandy, soil samples classified as buried OC peaks had significantly higher estimated clay contents ($p = 0.05$) than those that were not classified as peaks.

In general, the floodplain-stratified sites in the MF Snoqualmie stored higher densities of OC than the Big Sandy (Figure 2a, b). Figure 2a includes zero values (i.e., sites with no OC-bearing sediment, only present in the MF Snoqualmie slope-stratified group), whereas Figure 2b does not, because sample units in Figure 2b are individual soil samples. Comparing these two groups, it appears that soils in the MF Snoqualmie can exhibit much higher OC concentrations than those in the Big Sandy, but in general, there are many more reaches with no fine sediment

available to store OC in the MF Snoqualmie.

At the scale of individual soil samples, we found that the depth below ground surface was by far the dominant control on OC concentration across all modeling groups. We used a cube root transform for all three mixed effects models of OC concentration. For MF Snoqualmie sites stratified by slope, deeper soil samples contained less OC ($\beta = -0.0084 \pm 0.0042$), whereas soil samples at higher elevations tended to contain more OC ($\beta =$

$0.0010 \pm 0.00099$). Depth was the only significant predictor of OC content for both MF Snoqualmie soil samples stratified by floodplain type ($\beta = -0.0084 \pm 0.0042$) and soil samples in the Big Sandy ($\beta = -0.0037 \pm 0.0019$).

Modeling MF Snoqualmie slope-stratified sites at the reach-scale, we found that moisture ($\beta = 0.0078 \pm 0.0031$), and whether the reach was unconfined ($\beta = 0.77 \pm 0.49$) control soil OC (cube root transformation, model adjusted $R^2 = 0.54$, $p < 0.0001$). Modeling MF Snoqualmie floodplain-stratified sites at the site scale, we found that

canopy cover ($\beta = 0.012 \pm 0.011$) and moisture ($\beta = 0.0040 \pm 0.0011$) are controls on soil OC (cube root transformation, model adjusted $R^2 = 0.67$, $p < 0.0001$). Modeling Big sandy sites at the reach scale, we found that



soil depth (β = -0.012 ± 0.0071) and moisture (β = 0.014 ± 0.0026) are dominant controls on soil OC concentration (no transformation, model adjusted $R^2$ = 0.69, p < 0.0001).

In general, moister, deeper soils store more OC at the reach scale, whereas OC tends to vary dominantly with depth at the scale of individual soil samples. Although estimated clay content did not emerge as a significant predictor of OC concentration, it is used to calculate clay-held water to correct our LOI-based OC concentration measurements, making it important in determining OC for each sample.

### 3.2 Soil Texture

In general, soil texture followed a predictable trend with river size between model groups (Figure 2d). Floodplain type-stratified sites in the MF Snoqualmie stored the most clay, followed by slope-stratified sites and then sites in the Big Sandy.

Modeling soil texture at the individual soil sample scale across slope-stratified sites in the MF Snoqualmie, we found whether the reach was confined (β = 5.42 ± 5.32), and whether the bed material was dominantly sand (β = 10.47± 6.13) to be dominant controls on estimated clay content. Modeling soil texture for sites stratified by floodplain type in the MF Snoqualmie yielded no significant trends. In the Big Sandy, we found that either valley width (β = 0.0050± 0.0032) or whether the stream was unconfined (β = 0.41± 0.33) as well as depth below ground surface (β = -0.0069 ± 0.0033 for model with valley width but not confinement) significantly control soil texture.

To summarize, sites from unconfined, lower energy reaches in the MF Snoqualmie and sites from reaches with wider valley bottoms and at lower depths in the Big Sandy exhibited finer soils.

### 3.3 Soil Moisture

Soil moisture was less variable between basins than either texture or OC concentration (Figure 2c). All model groups exhibited similar soil moisture conditions, although there was significant variability within each model group.

Soil moisture at MF Snoqualmie sites stratified by slope is dominantly controlled by channel slope (β = -13.15± 9.11), elevation (β = 0.0046 ± 0.0038), and whether the stream is unconfined (β = 3.89 ± 2.67; model adjusted $R^2$ = 0.38, p < 0.0001). At MF Snoqualmie sites stratified by floodplain type, estimated clay content (β = 0.060 ± 0.042) and whether the floodplain unit had standing water (β = 1.15 ± 0.71) significantly controlled soil moisture (model adjusted $R^2$ = 0.46, p < 0.0001). In the Big Sandy, soil depth (β = 0.012 ± 0.012), elevation (β = 0.0023 ± 0.0014), and whether the reach was unconfined (β = 0.91 ± 0.75) significantly controlled soil moisture (model adjusted $R^2$ = 0.35, p < 0.0001).

### 4. Discussion

#### 4.1 Understanding Spatial Variability in OC Concentration in Floodplain Soils (H1 and H2)





Comparing the MF Snoqualmie to the Big Sandy shows that the wetter, higher primary productivity basin is capable of storing greater concentrations of OC in floodplain soils, but that both regions generally store similar OC concentrations in floodplain soils. This result partially agrees with the examination of subalpine lake deltas by Scott and Wohl (2017). In that study, subalpine lake deltas in the MF Snoqualmie were compared to deltas in the

drier Colorado Front Range. Subalpine lake deltas displayed similar OC concentrations, likely due to competing but complementary OC stabilization and respiration mechanisms in each region. Those deltas represent a subset of the broader valley bottom soils studied here. This more expansive study points to both geomorphic controls, such as valley bottom geometry, and factors influenced by climate, such as canopy cover, as controls on OC storage in valley bottoms. These results agree with the results of Lininger et al. (2018),which indicate that geomorphic context

and vegetation dynamics control OC concentration on floodplain soils along large, lowland rivers in Alaska, USA.

At the reach or site scale, wetter soil profiles consistently yielded higher OC concentrations in all model groups. However, moisture did not differ significantly between model groups (Figure 2c), indicating that this alone cannot explain differences between basins. Soils tend to be finer in the MF Snoqualmie, but clay content is not an important predictor of OC concentration in studied soils. Although clay content likely influences OC concentration

based on previous research (Hoffmann et al., 2009), the inclusion of coarse soil material (including particulate organic matter) in our samples may explain the lack of an observed correlation here. Although confinement plays a strong role in determining OC concentration in MF Snoqualmie sites stratified by slope, it doesn't differ significantly between basins (52% of Big Sandy reaches are unconfined compared to 63% of MF Snoqualmie reaches). The major differences between these basins are their hydroclimatic and disturbance regimes. The MF

Snoqualmie is at a lower elevation, is wetter, and has denser and higher biomass forests (Smithwick et al., 2002), compared to the drier, sparser parkland forests of the Big Sandy, which likely also experiences more frequent fires based on fire histories of nearby regions (recurrence interval on the order of $10^1$ - $10^2$ yrs; Houston, 1973; Loope and Gruell, 1973).

Between basins, it is likely that hydroclimatic regime, influencing primary production, plays some role in

the MF Snoqualmie's higher maximum OC concentrations in floodplain soils compared to those of the Big Sandy. However, smaller-scale factors such as soil texture and moisture also likely play a role and are not related to drainage area (Table 1), indicating that neither OC concentration nor its controlling factors vary continuously along a river network, and thus supporting H2a and H2b. This also indicates that local factors, set largely by geomorphic and groundwater dynamics, play a significant role in modulating the effect of climate on OC concentrations. If the

MF Snoqualmie and Big Sandy displayed significantly different OC concentrations, our first hypothesis regarding the inter-basin controls on OC concentration would be supported. However, we instead find that climate and primary productivity only partially determine OC concentrations, especially when viewed in the context of geomorphic and hydrologic variability. Thus, the results do not support H1.

Each basin (or model group) is slightly different in terms of the controls on soil OC concentration,

moisture, and texture. In the MF Snoqualmie sites stratified by slope, higher elevation sites displayed higher OC concentrations. This is contrary to the general trend in primary productivity, which decreases with increasing elevation. However, it is important to note that the headwaters of the MF Snoqualmie are dominated by lakes, deltas,



and other depositional features in relatively broad, glacially carved valleys. Subalpine lake deltas have been shown to store high OC concentrations in this basin (Scott and Wohl, 2017), and many of the highest OC concentrations we measured were located in broad, wet meadows, subalpine lake deltas, or other unconfined, high elevation reaches. Such unconfined sites likely have significantly cooler temperatures and tend to have higher soil moisture contents,

as shown by our modeling (Table 1). As such, although high elevation MF Snoqualmie sites may receive less OC input, they likely have a low rate of OC respiration, resulting in higher OC concentrations on the whole, which agrees with the result of Bao et al. (2017). In the Big Sandy, our modeling suggests that the lower temperatures and higher moisture (Table 1) at higher elevations do not compensate for the lower primary productivity, as elevation does not correlate to OC concentration.

10       In both basins, unconfined reaches contained wetter and finer textured soils, which may result in a higher soil OC capacity. Although confinement only related directly to OC content in MF Snoqualmie sites, it does play a strong role in determining moisture, which plays a role in regulating OC concentration in both basins, likely via inhibiting microbial activity (Howard and Howard, 1993). The relevance of channel slope in determining soil moisture in the MF Snoqualmie but not Big Sandy may reflect the prevalence of high-gradient, debris-flow

dominated channels in the MF Snoqualmie that largely exhibited only gravel to boulder substrate, which we assume stores minimal fine sediment, moisture, or OC.

      In the Big Sandy, higher soil depths were related to more moisture and finer texture, but less OC concentration. This indicates the trend in OC with depth likely dominates the signal of OC concentration, with deeper sites containing a higher proportion of OC-depleted, deep samples.

## 4.2 Inferring Sources of OC to Floodplain Soils (H3)

      OC can be input to floodplain soils by two primary mechanisms. First, dissolved and particulate OC can be deposited on floodplain surfaces by overbank sediment deposition, thus integrating fluvial sedimentary OC into the

floodplain soil profile. Second, litter and decomposing vegetation on the floodplain surface, in addition to decomposing wood that may have been deposited by overbank flows, can input OC to floodplain soil.

      Our modeling of OC concentration yielded results consistent with previous investigations of controls on soil OC storage capacity (Jobbágy et al., 2000; Sutfin and Wohl, 2017). Sites in the heterogeneous floodplain of the MF Snoqualmie displayed a direct correlation between canopy cover and OC concentration, indicating that

increased litter inputs lead to increased floodplain soil OC concentration. Sediment inputs likely differ between floodplain depositional units (e.g., coarser sediment may deposit on point bars compared to filled secondary channels), which were not found to be an important predictor of OC concentration. This indicates that vegetation inputs may be more dominant than fluvial sediment inputs at these sites.

      The finding that buried OC peaks in the MF Snoqualmie do not have abnormally high clay contents

supports the interpretation that wood and litter inputs to soil are the dominant source of OC in the floodplain soils we examined. Buried peaks can be either layers created by overbank deposition and subsequent burial of fine, OC-bearing sediments (Blazejewski et al., 2009; Ricker et al., 2013), buried pieces of wood (Wohl, 2013), or buried





organic horizons that are now capped by sediments that prevent OC respiration. If overbank deposition of fine sediment caused OC peaks, we would expect to see the soil samples classified as peaks exhibiting high clay contents, indicating finer sediment. Instead, our results suggest that in the MF Snoqualmie, buried peaks are likely the result of either buried organic horizons or buried wood. We observed large pieces of decaying, buried wood in

floodplain cut banks in the MF Snoqualmie, supporting this inference.

In the Big Sandy, the two cores that exhibited peaks were collected from the same meadow, just downstream of a now-filled former lake that is a potential source of fine sediment. The channels draining this meadow exhibit an anabranching planform, indicating the potential to deposit and bury packets of potentially OC-rich, fine sediments. However, the majority of cores did not exhibit OC peaks, indicating OC input mainly from

vegetation at the surface and continuing OC respiration at depth.

OC variation within each core is dominantly a function of depth. We observe a negative correlation between depth below ground surface and OC concentration, which has been observed in other studies, including mountain wetlands and floodplains (Jobbágy et al., 2000; Scott and Wohl, 2017; Sutfin and Wohl, 2017; Zhao et al., 2017). In general, this indicates that OC is enriched at the surface and decomposes with depth. This fits with our

finding that the majority of our cores do not exhibit significant OC peaks at depth and supports the dominance of litter and wood OC inputs to floodplain soils. These results support our hypothesis that decaying litter and wood, not overbank sediment deposition, dominates the input of OC to floodplain soils in our study basins (H3). Other basins that experience overbank flows, accompanying deposition of fine sediment, and burial of organic layers exhibit OC storage that is likely dominated by fluvial deposition (e.g., Blazejewski et al., 2009; D'Elia et al., 2017; Ricker et al.,

2013). Thus, it is likely that flow regime, lateral connectivity, and sediment transport dynamics regulate whether floodplain soil OC is dominantly input by overbank deposition of fine material or decaying litter and wood.

**4.3 Conceptual Model of Soil OC Concentration in Floodplain Soils**

We present a conceptual model to summarize our results and place them in the context of recent work examining the controls on OC storage in soils (Figure 3). OC is input to floodplains either through the decay of vegetation or the deposition of fine, OC-rich sediment. This input of OC only determines OC concentrations insofar as floodplain soils are capable of storing OC. That storage is effectively determined by a balance between processes that remove OC from floodplains, namely respiration or erosion followed by respiration (Berhe et al., 2007), and

processes that regulate OC availability to microbes, namely the capability of the mineral fraction of the soil to sorb OC.

OC sorption capacity reflects a few specific processes. Although soil texture generally relates to the ability of OC to sorb to mineral grains and resulting OC availability, soil chemistry also plays a strong (and potentially dominant) role in regulating OC sorption capacity (Rasmussen et al., 2018). Soil texture is largely determined by

valley morphology, according to our modeling (Table 1), placing valley morphology and resulting sediment transport dynamics (Gran and Czuba, 2017; Wohl et al., 2017b) as an indirect control on sorption capacity.





Respiration rate is largely determined by microbial activity and the availability of OC to microbes. Erosion can rapidly expose soil OC to microbial respiration (Berhe et al., 2007), whereas soils that reside in largely anoxic conditions can exhibit low rates of microbial respiration (Boye et al., 2017). Our results suggesting that moisture controls OC content support the idea that drier soils likely have higher rates of microbial respiration of OC. Moisture is a function of texture, valley bottom morphology, and elevation (a proxy temperature) in our modeling (Table 1). Comparing floodplain types in the MF Snoqualmie, we find that types with standing water exhibit significantly higher soil moisture contents than those without standing water. This indicates spatial variability in moisture content and likely microbial activity (Howard and Howard, 1993). In our modeling, this effect translates to spatial variability in OC concentration within floodplains and across entire basins.

To summarize, we propose that OC inputs are regulated by the capacity of soils to store OC and suppress microbial respiration, allowing OC to accumulate. OC inputs to floodplain soils come from either autochthonous litter accumulation on the floodplain surface, allochthonous wood deposition, or allochthonous deposition of fine, OC-bearing sediments. In these systems, deposition of fine material in overbank flows is rare, leading us to infer that autochthonous litter and allochthonous wood inputs to floodplains dominate OC input. Where soils are more moist, microbial respiration is inhibited and more OC is stored. Although soil texture is likely not a limiting factor on OC concentration in these floodplains, finer textured soils likely have a higher sorption capacity, retaining more of the OC input from decaying plant material.

## 5. Conclusion

We present floodplain soil OC concentration data from two disparate watersheds to compare how inter-basin variability between the two watersheds compares with intra-basin variability in geomorphic and hydrologic characteristics in determining OC concentration. Our results indicate that OC concentration in mountain floodplain soils does not vary predictably along a longitudinal gradient, nor does it vary substantially between basins with differing climatic and vegetation characteristics. Instead, geomorphic and hydrologic characteristics, such as valley bottom morphology and soil moisture, dominantly determine floodplain OC concentration.

In our study basins, decaying litter and wood, and not overbank deposition of fine, OC-bearing sediment, is the main source of OC to floodplain soils. It is unclear whether that decaying vegetation is dominated by autochthonous litter inputs or transported downed wood. In comparing our basin to other studied floodplain soils, it seems that vegetation dynamics play a strong role in determining OC concentrations when fine sediment is not regularly deposited on floodplain surfaces. However, we suggest that floodplain soil characteristics, set by geomorphic and hydrologic conditions, regulate how OC inputs translate to the spatial distribution of OC along a river network.

This implies that OC storage in floodplains likely cannot be predicted using consistent, downstream trends, and that management prioritization designed to facilitate floodplain OC storage should be based on local geomorphic and hydrologic process variability within each basin. For instance, management to increase OC sequestration in floodplain soils will likely be more effective where floodplains are unconfined and soils already experience high moisture conditions for much of the year. Along these lines, our results show that modeling the

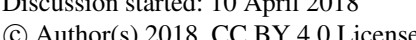



floodplain biospheric OC pool to predict its response to warming and subsequent effects on climate based on regional factors such as climate and net primary productivity likely misses the substantial inter-basin variability in OC concentration and storage resulting from variability in valley bottom geometry and both geomorphic and hydrologic processes (e.g., Doetterl et al., 2015).

Although our results provide some insights, the question of whether OC stored in floodplain soil comes dominantly from allochthonous versus autochthonous sources remains open. Our results imply that more productive, spatially heterogeneous floodplains likely input more OC to soils. Floodplain OC concentration, while mediated largely by moisture dynamics, likely depends mainly on OC inputs from productive riparian forests. This implies that management of OC storage in mountain river floodplains should focus on the restoration of riparian zones to

maintain OC input to soil (e.g., Bullinger-Weber et al., 2014). More detailed studies in regions with varying sediment transport and hydrologic regimes are needed to determine what conditions favor autochthonous versus allochthonous OC inputs, but our results suggest that autochthonous sources dominate floodplain OC storage in basins with relatively low rates of vertical accretion and high channel-floodplain connectivity that promotes floodplain wetlands.

**Acknowledgements**

This work was funded by NSF grant EAR-1562713. We thank Ellen Daugherty for extensive assistance in field work and Katherine Lininger for stimulating discussion that improved the manuscript. All data supporting the analyses presented here can be found in Table S1.

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





**Figures**

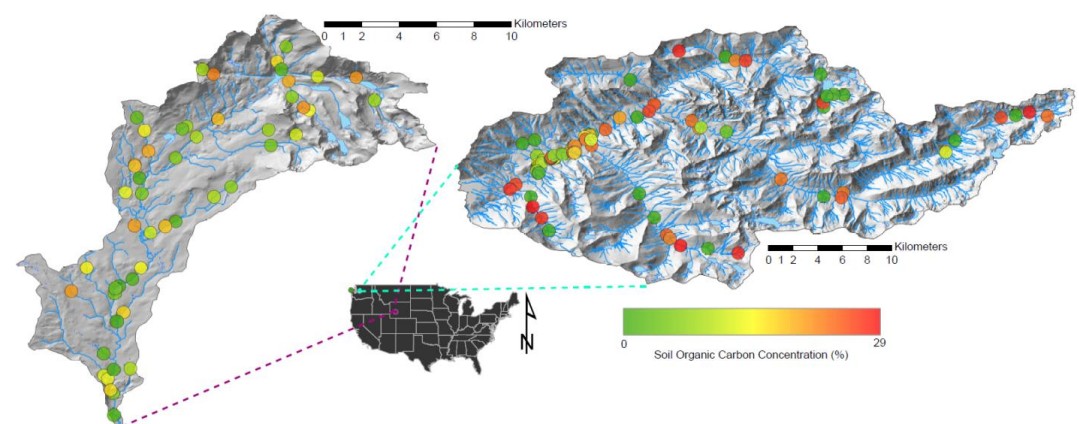

5    **Figure 1: Map showing the location, topography, sampling sites, and stream network of the sampled basins.
Big Sandy, Wyoming, on left and MF Snoqualmie, Washington, on right. Circles represent sampling locations
at which floodplain soil OC was measured. Sample sites are colored by OC concentration.**



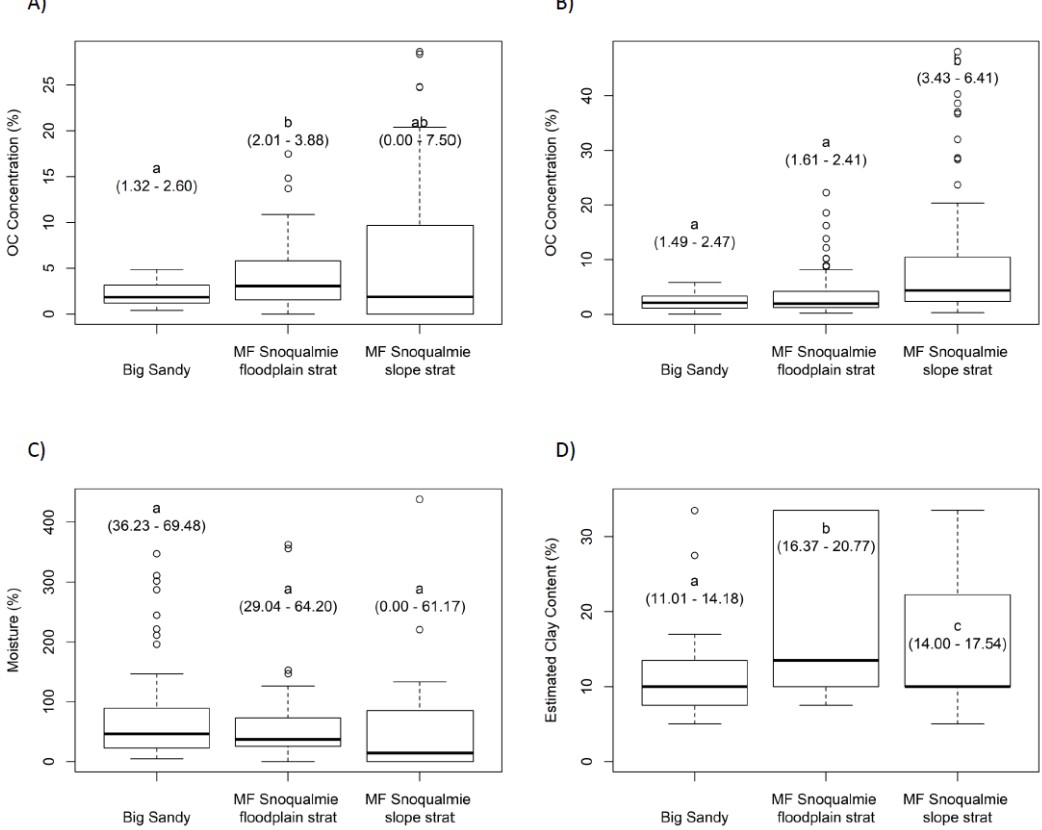

**Figure 2: Boxplots showing comparisons between model groups of OC concentration at the reach-scale (A), OC concentration at the scale of individual soil samples (B), moisture at the reach-scale (C), and estimated clay content at the scale of individual soil samples (D). Ends of dotted lines represent 1.5 times the inter-**

5 **quartile range, which is represented by boxes. Bold line represents median. Circles represent outliers. Letters indicate probable differences between groups based on pairwise Wilcoxon (A-C) or t tests (D) with a holm correction. Ranges in parentheses below letters show the 95% confidence interval on the median value for the group (A-C) or the mean value for the group (D) where median confidence intervals were overly constrained due to the categorical nature of our estimated clay content data.**





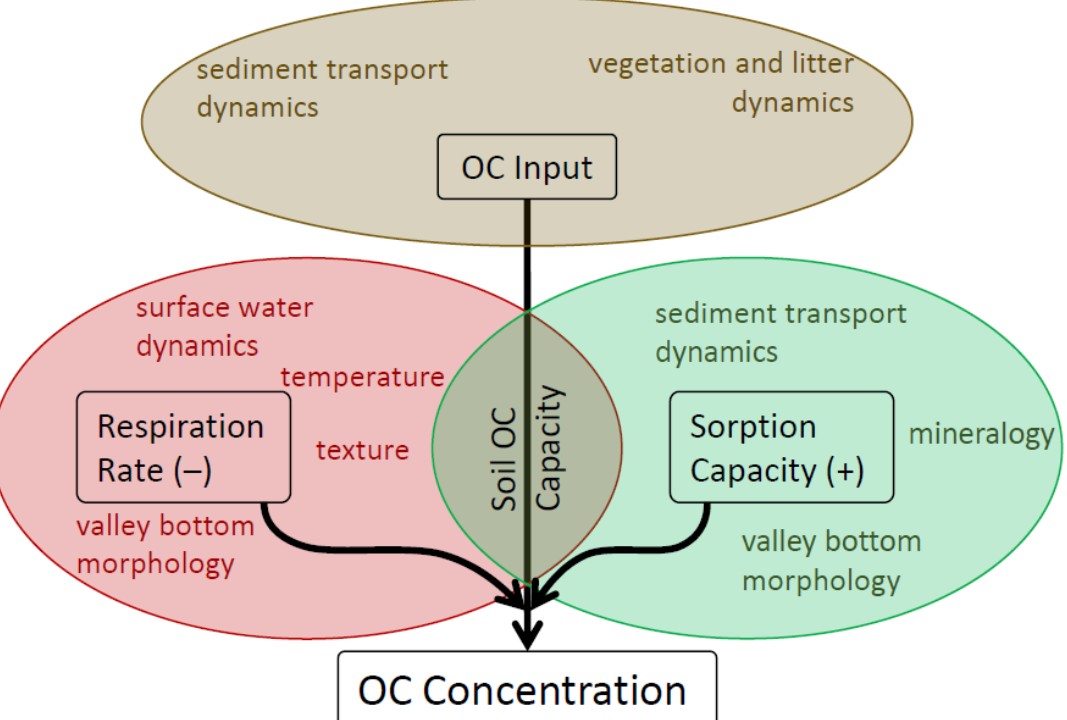

**Figure 3: Conceptual model of physical processes that influence OC concentration in floodplain soils. Each ellipse corresponds to a major factor that influences OC concentration. Colored text within each ellipse denotes factors that influence OC inputs, sorption capacity, or respiration rate. As sorption capacity increases, so does the OC capacity of the soil. Conversely, as respiration rate (or likelihood) increases, the soil OC capacity decreases. Floodplain soils can only develop high concentrations of OC if there are high rates of OC input. However, the capacity of the soils to store OC regulates that input, and is determined by the competing influences of sorption capacity and respiration rate. See text for further details.**



**Tables**

**Table 1: Matrix of all models presented in text. Each model is listed by model group, response variable, and scale. Scale refers to the sample unit of the model, where site refers to a core, with the response averaged over all the individual soil samples in the core. For each variable and model, grey fill indicates that the variable was included in either model selection or the full mixed-effects model. A minus (−) indicates that the variable was selected as important in predicting the response, and denotes an indirect correlation, whereas a plus (+) indicates a direct correlation. In the case of confinement, a plus indicates that unconfined streams display a higher magnitude response variable. In the case of bed material, a plus indicates that samples with sand exhibit a higher value of the response. NA indicates that either the variable wasn't measured for that basin or model group or that it is the model response.**





**Variables**

| Model Group | Response | Scale (Sample Unit) | Confinement | Bedform | Channel Slope | Bed Material | Multithread | Valley Width | Bankfull Width | Bankfull Depth | Stream Power | Floodplain Type | Standing Water | Depth¹ | Clay Content | Moisture | Logging Nearby | Grasses Present | Shrubs Present | Trees Present | Elevation | Basin Slope | Canopy Cover | NLCD | Drainage Area |
|---|---|---|---|---|---|---|---|---|---|---|---|---|---|---|---|---|---|---|---|---|---|---|---|---|---|
| MF Snoqualmie stratified by slope | OC (%) | Soil Sample | | | | | | | | | | NA | NA | − | | | | | | | + | | | | |
| | OC (%) | Site | + | | | | | | | | | NA | NA | | | + | | | | | | | | | |
| | Moisture (%) | Site | + | | − | | | | | | | NA | NA | | NA | NA | | | | | + | | | | |
| | Texture (%) | Soil Sample | + | | | + | | | | | | NA | NA | − | NA | NA | | | | | | | | | |
| MF Snoqualmie stratified by floodplain type | OC (%) | Soil Sample | NA | NA | NA | NA | NA | NA | NA | NA | NA | | | − | | | | | | | | | | |
| | OC (%) | Site | NA | NA | NA | NA | NA | NA | NA | NA | NA | | | | + | + | | | | | | | + | | |
| | Moisture (%) | Site | NA | NA | NA | NA | NA | NA | NA | NA | NA | | + | − | NA | NA | | | | | | | | | |
| | Texture² (%) | Soil Sample | NA | NA | NA | NA | NA | NA | NA | NA | NA | | | + | NA | | | | | | | | | | |
| Big Sandy | OC (%) | Soil Sample | | | | | NA | NA | NA | NA | NA | NA | NA | − | | NA | NA | NA | NA | NA | | | | | |
| | OC (%) | Site | | | | | NA | NA | NA | NA | NA | NA | NA | − | | + | NA | NA | NA | NA | + | | | | |
| | Moisture (%) | Site | + | | | | NA | NA | NA | NA | NA | NA | NA | + | | NA | NA | NA | NA | NA | | | | | |
| | Texture³ (%) | Soil Sample | + | | | | NA | NA | + | NA | NA | NA | NA | − | NA | NA | NA | NA | NA | NA | | | | | |

¹ Depth refers to either the soil sample depth below the ground or the total depth of the core, depending on the sample unit.
² No significant results were observed for this model.
³ For this model, both valley width and confinement predict texture and can be interpreted interchangeably. However, including both in the same model would yield problems due to multicollinearity.