# Peer review of "Geomorphic regulation of floodplain soil organic carbon concentration in watersheds of the Rocky and Cascade Mountains, USA"

_Earth Surface Dynamics, 2018_

## Referee Comment (RC1) · Anonymous Referee #1 · 8 Jun 2018

Scott and Wohl present an analysis and interpretation of a dataset containing soil organic matter concentrations on floodplain soils. Due to the dynamic alluvial nature, these soils often have large carbon stores and function differently than soils on hill slopes. Scott and Wohl have made a valuable step in understanding the geomorphic factors controlling the distribution of organic matter in these profiles. This contribution will be important to the understanding of how carbon is mobilized and transported to long-term depositional areas. I thank the authors for a very well written manuscript and I have only a few editorial comments. I have a few moderately important suggestions that the authors should consider and respond to: 1. The redox condition of soils is hugely important in determining the fate of associated carbon (as noted by your dis-

cussion on soil moisture and respiration). This has been recently been brought to the fore by Keiluweit, et al. (2018). Changing redox condition in these soil profiles may be really important for destabilizing mineral associated carbon. But probably most important was that much of the deposited carbon was probably fixed in an "oxic" upland location - so that the stabilizing minerals (Fe) could be destabilized in the floodplain. The redoximorphioc conditions of floodplain soils are complex since they it is more than just a simple function of depth and moisture since hyporheic flow can bring DO into portions of the profile and floodplain that may seem distal to the river. Soil morphological characterization including characterizing the mottling, gleying, and soil color could have helped with those hypotheses that moisture was important in destabilizing or preserving soil carbon (or that dryer sites have higher respiration). Where these sorts of observations made and recorded in a systematic way so that they could be included in the analysis? At any rate I think this emerging topic in soil carbon stabilization needs to be incorporated into the discussion. 2. How does the dominant precipitation type affect erosion and sedimentation in a watershed? The Snoqualimie River is likely more dominated by rain than the Big Sandy which is likely more dominated by snow. Will this have an impact on what is mobilized by the river network (i.e. source of POM)? If the authors think so, then I suggest these topics be discussed. 3. I would really like to see a figure summarizing the profile OC concentrations. I realize it could be complex. Something that shows means by depth (box plots? with actual data points in a lighter shade?). This would go a long way in illustrating that depth is the primary controller of OC concentrations in these profiles. 4. Soils in the Snoqualmie basin are dominated by Andisols and Spodosols which have a disproportionate ability to stabilize soil carbon (especially relative to the Big Sandy). Can you please discuss how the soil type might have affected the differences between the basins?

Other than those three comments: Page 11 Line 25: This took me a couple times to figure out. Deposition of POM on the surface of the soil would be included in the first mechanism of this paragraph. I think you're trying to distinguish it from terrestrial processes (authoncthonous inputs) and/or include it at the same time in this paragraph.

You might be able to clear this up by dealing with the river deposition of allocthonous POM as a sort of confounding factor later in the discussion (or not).

Page 12 Line 5 (and throughout): I have mainly seen that O-horizon (litter and duff) floats away when floodplains are flooded. This material tends to be trapped downstream (e.g. log jams) and maybe it's these areas that are buried and able to preserve litter.

References cited: Keiluweit M, Wanzek T, Kleber M, Nico P, Fendorf S. Anaerobic microsites have an unaccounted role in soil carbon stabilization. Nature communications. 2017 Nov 24;8(1):1771.

---

## Author Comment (AC1) · 11 Jul 2018

**Author's Response to Reviewer 1**

We are grateful for the constructive and detailed commentary provided by the reviewer. We have revised the manuscript to address the reviewer comments and responded to those comments below in *italics*.

Anonymous Referee 1 Comments:

Scott and Wohl present an analysis and interpretation of a dataset containing soil organic matter concentrations on floodplain soils. Due to the dynamic alluvial nature, these soils often have
large carbon stores and function differently than soils on hill slopes. Scott and Wohl have made a valuable step in understanding the geomorphic factors controlling the distribution of organic matter in these profiles. This contribution will be important to the understanding of how carbon is mobilized and transported to long-term depositional areas. I thank the authors for a very well written manuscript and I have only a few editorial comments. I have a few moderately important
suggestions that the authors should consider and respond to:

*Thank you. We appreciate your comments, especially in bringing up new avenues of thought for considering our results.*

1. The redox condition of soils is hugely important in determining the fate of associated carbon (as noted by your discussion on soil moisture and respiration). This has been recently been
brought to the fore by Keiluweit, et al. (2018). Changing redox condition in these soil profiles may be really important for destabilizing mineral associated carbon. But probably most important was that much of the deposited carbon was probably fixed in an "oxic" upland location - so that the stabilizing minerals (Fe) could be destabilized in the floodplain. The redoximorphioc conditions of floodplain soils are complex since they it is more than just a simple function of
depth and moisture since hyporheic flow can bring DO into portions of the profile and floodplain that may seem distal to the river. Soil morphological characterization including characterizing the mottling, gleying, and soil color could have helped with those hypotheses that moisture was important in destabilizing or preserving soil carbon (or that dryer sites have higher respiration). Where these sorts of observations made and recorded in a systematic way so that they could be
included in the analysis? At any rate I think this emerging topic in soil carbon stabilization needs to be incorporated into the discussion.

*We appreciate the suggested reference and this new avenue of discussion. We unfortunately did not systematically record soil morphological characterization in the field, and as such can't directly speak to the impacts of variability in redoximorphic conditions on our results. However,*
*after reading the reference you suggest and a couple others, we agree that this topic should be noted in our discussion.*

*We have added a few sentences to the 2nd paragraph of section 4.1, added a clause to a sentence in section 4.3, and added "redox condition" to the factors listed as controlling respiration rate in our conceptual model to discuss this topic.*

2. How does the dominant precipitation type affect erosion and sedimentation in a watershed? The Snoqualimie River is likely more dominated by rain than the Big Sandy which is likely more dominated by snow. Will this have an impact on what is mobilized by the river network (i.e. source of POM)? If the authors think so, then I suggest these topics be discussed.

*This isn't something we previously considered, and we appreciate you bringing it to our*
*attention. We have reviewed the literature we are aware of on this topic (comparing snowmelt vs rain driven flows in terms of OM flux) and determined that there is insufficient evidence from that literature to make supported inferences regarding the potential effects of the variability in precipitation regime between these two basins on their resulting OC transport. While we acknowledge that precipitation regime likely has some effect on OM flux, we are too uncertain of*
*the nature of that effect to appropriately discuss it here. In addition, our results indicate that in-situ OC production likely dominates the signals we observe in these floodplains. As such, we are unsure how precipitation regime might influence our results or interpretation, and we have thus not added a discussion of this topic to the manuscript.*

*For reference, the papers we reviewed are (as DOIs):*

*10.1007/s10533-009-9401-1*

*10.1007/s10533-011-9589-8*

*10.1007/s10533-010-9416-7*

*10.1002/esp.1882*

3. I would really like to see a figure summarizing the profile OC concentrations. I realize it could
be complex. Something that shows means by depth (box plots? with actual data points in a lighter shade?). This would go a long way in illustrating that depth is the primary controller of OC concentrations in these profiles.

*Thanks for the suggestion. We like the idea of doing box plots with data points shown simultaneously. While we unfortunately were unable to plot profiles (i.e., line plots*
*corresponding to each profile) in a way that was readable on a single chart, we have now made a figure that has boxplots showing OC concentrations binned in average depths, with points shown transparently for each bin. The only drawback is that the points aren't connected to group them by profile. However, we think this still provides a decent portrayal of the general trends of OC concentration with depth in our samples. We have added this figure to the results section and*
*used it to provide a better topic sentence for the first paragraph in section 3.1.*

4. Soils in the Snoqualmie basin are dominated by Andisols and Spodosols which have a disproportionate ability to stabilize soil carbon (especially relative to the Big Sandy). Can you please discuss how the soil type might have affected the differences between the basins?

*Good point, thanks for bringing this up. We have chosen to integrate this idea in a few places, namely by citing a review on C storage in andic soils. First, we discuss this as a potential reason for our observation of higher maximum OC concentrations in the MF Snoqualmie. Second, we have added the potential effects of soil minerology and chemistry on microbial respiration to our conceptual model (section 4.3)*

Other than those three comments:

Page 11 Line 25: This took me a couple times to figure out. Deposition of POM on the surface of the soil would be included in the first mechanism of this paragraph. I think you're trying to distinguish it from terrestrial processes (authoncthonous inputs) and/or include it at the same time in this paragraph. You might be able to clear this up by dealing with the river deposition of allocthonous POM as a sort of confounding factor later in the discussion (or not).

*We see how this could be confusing. We were trying to separate OC inputs into either fine sediment of FPOM deposited by overbank flows and coarser OM deposited as litter or large wood. Re-reading this, we realize this distinction does not come across clearly, so we have instead organized these two categories by whether the input is autochthonous or allochthonous to improve clarity.*

Page 12 Line 5 (and throughout): I have mainly seen that O-horizon (litter and duff) floats away when floodplains are flooded. This material tends to be trapped downstream (e.g. log jams) and maybe it's these areas that are buried and able to preserve litter.

*We have now acknowledged this as a potential interaction between large wood and fine organic matter. However, we felt it was more appropriate to acknowledge this later in this section when summarizing our interpretations.*

[revised manuscript text omitted]

---

## Referee Comment (RC2) · Anonymous Referee #2 · 9 Aug 2018

"Geomorphic regulation of floodplain soil organic carbon concentration in watersheds of the Rocky and Cascade Mountains, USA" by Scott and Wohl uses data from distinct basins to identify key drivers of inter- and intra-basin organic carbon (OC) storage in soils. While we often focus on rivers as exporters and transformers of carbon, we rarely acknowledge the potential for river corridors to store organic matter. This paper does a nice job of highlighting the different controls on this storage function, is presented clearly, and will be of great interest to a diverse group of researchers and managers. Below I offer a few major and minor comments to highlight the novelty and contributions of this work.

[Figure]

Major comments

1. Highlight novelty. As written, the authors largely omit much-needed reminders of how and why this work is novel, and which key knowledge gaps this paper addresses. The abstract does a modest job of this, but the introduction and discussion do not sufficiently highlight the "unknowns". While many of these controls on OC input and storage are intuitive, datasets like the one presented here are relatively rare, as are the studies that are able to tease apart the relative importance of expected controls across multiple sites.

2. Conceptual figure. I was very excited to see this, as I wrote "need for conceptual figure?" a page or two before Figure 3 was introduced in the text. However, the link between the findings in this paper and the take-home messages from the conceptual figure are unclear. If the authors find the Veg/Litter, Valley bottom morph, and Moisture (is "surface water dynamics" = "moisture" in the figure? Clarify) are dominant controls, why aren't they highlighted in the figure? I realize this study, while very thorough, was only in 2 basins and the authors may not want to exclude the possibility of other primary controls in other places, but, related to #1 above, including this figure at the end of the discussion should highlight how this work has changed or improved our understanding.

3. Floodplains and the aquatic/terrestrial limbo. In 1.1, the authors acknowledge the uncertainty of "whether OC concentration follows a trend similar to uplands…." but do not revisit this. As someone who feels that river corridors/floodplains get left out of these discussions too often by surface water or upland studies, I urge the authors to come back to this point in the discussion. Is this assumption of a similar trend a reasonable one?

Minor comments as page,line

1,8 – Consider making this opening statement more active: "mountain rivers have the potential for high organic carbon (OC) storage by retaining…."

1,10 – Here and elsewhere (e.g. 7,35 and other sections). Why present tense? The authors often switch back and forth in a single paragraph. Keep consistent. Past sense seems more appropriate for work already done.

2,35 – I enjoyed thinking about allochthonous/autochthonous from the floodplain perspective. Thanks for coming back to this in the discussion (14,5) as well!

4,1 – Access roads and trails here as well, no? I know the winds can be pretty remote, but it's nice to get a feeling for direct human impacts as mentioned for logging in the MF Snoqualmie basin (even if they are minimal).

9,3 – I really appreciated these summaries at the end of each paragraph. Great way to stay in results mode but not loose sight of overall trends.

10,6 – Respiration is a mechanism. Consider restating. Perhaps "stabilization and loss mechanisms"?

10,9 – Missing a space: "2018),which"

13,10 – As written, could be read as inputs are regulated by storage. Could restate as "the fate of OC inputs are regulated..."

19, Figure 1 – coordinates?

21, Figure 3 – see major comment #2 above.

---

## Author Comment (AC2) · 20 Sep 2018

**Author's Response to Reviewer 2**

We are grateful for the constructive and detailed commentary provided by the reviewer. We have revised the manuscript to address the reviewer comments and responded to those comments below in *italics*. The revised manuscript is shown below our response to reviewer comments

Anonymous Referee 2 Comments:

"Geomorphic regulation of floodplain soil organic carbon concentration in watersheds of the Rocky and Cascade Mountains, USA" by Scott and Wohl uses data from distinct basins to identify key drivers of inter- and intra-basin organic carbon (OC) storage in soils. While we often focus on rivers as exporters and transformers of carbon, we rarely acknowledge the potential for river corridors to store organic matter. This paper does a nice job of highlighting the different controls on this storage function, is presented clearly, and will be of great interest to a diverse group of researchers and managers. Below I offer a few major and minor comments to highlight the novelty and contributions of this work.

*We are grateful for your comments and feel that the resulting revisions have improved the manuscript.*

Major comments

1. Highlight novelty. As written, the authors largely omit much-needed reminders of how and why this work is novel, and which key knowledge gaps this paper addresses. The abstract does a modest job of this, but the introduction and discussion do not sufficiently highlight the "unknowns". While many of these controls on OC input and storage are intuitive, datasets like the one presented here are relatively rare, as are the studies that are able to tease apart the relative importance of expected controls across multiple sites.

*We have revised the introduction to better highlight the knowledge gaps that this study helps to address, specifically with regard to the spatial distribution of OC in mountain river networks, the source of that OC, and the multi-scale controls on OC concentration. We also note that the first paragraph of section 1.1 further elaborates on the research needs specific to understanding floodplain soil OC dynamics. Our revised introduction now provides an overview, including a review of relevant literature, that leads into the more detailed explanation of current knowledge gaps and how we address them in section 1.1.*

2. Conceptual figure. I was very excited to see this, as I wrote "need for conceptual figure?" a page or two before Figure 3 was introduced in the text. However, the link between the findings in this paper and the take-home messages from the conceptual figure are unclear. If the authors find the Veg/Litter, Valley bottom morph, and Moisture (is "surface water dynamics" = "moisture" in the figure? Clarify) are dominant controls, why aren't they highlighted in the figure? I realize this study, while very thorough, was only in 2 basins and the authors may not want to exclude the possibility of other primary controls in other places, but, related to #1 above, including this figure at the end of the discussion should highlight how this work has changed or improved our understanding.

*We have edited our conceptual model (now Figure 4) substantially based your comments and those of Reviewer 1. We have attempted to strike a balance between strictly presenting our results and generalizing the conceptual model to include results from previous work. However, we realize that it would be helpful to highlight how our results add to this conceptual model. As such, as have edited the text in each part of the figure to better correspond to the factors we actually modeled (soil texture, moisture, etc.). We also added language to the last paragraph in section 4.3 to better summarize how our results advance our understanding, specifically connecting geomorphic and hydrologic processes to OC storage in floodplain soils.*

3. Floodplains and the aquatic/terrestrial limbo. In 1.1, the authors acknowledge the uncertainty of "whether OC concentration follows a trend similar to uplands. . .." but do not revisit this. As someone who feels that river corridors/floodplains get left out of these discussions too often by surface water or upland studies, I urge the authors to come back to this point in the discussion. Is this assumption of a similar trend a reasonable one?

*We have revised the discussion to revisit this assumption more explicitly. In section 4.2, paragraph 5, we cite multiple previous studies to show that our finding of decreased OC concentration with depth is similar to other soil profiles. To address your comment, we have added more explicit language to this paragraph making it more clear that floodplains to exhibit trends in OC concentration with depth that are similar to uplands. Also, In addressing comments by reviewer 1, we have also added a figure (now Figure 3) showing OC concentration trends with depth. These revisions hopefully make it clearer to readers that mountain river floodplains do follow vertical soil OC concentration trends that are similar to uplands.*

Minor comments as page,line

1,8 – Consider making this opening statement more active: "mountain rivers have the potential for high organic carbon (OC) storage by retaining. . .."

*We have revised this sentence somewhat differently than this suggestion, but in the spirit of making it more impactful and active.*

1,10 – Here and elsewhere (e.g. 7,35 and other sections). Why present tense? The authors often switch back and forth in a single paragraph. Keep consistent. Past sense seems more appropriate for work already done.

*Our general framework with respect to verb tense is to use present tense in the abstract and introduction, past tense in the methods, and present tense in the results and discussion. We feel that present tense is appropriate in many places to discuss either things that we found that are assumedly still true in the present as well as things we actively do in the manuscript. We appreciate that you caught these inadvertent mistakes in verb tense consistency. We have revised each section to be of uniform verb tense, according to the framework outlined above.*

2,35 – I enjoyed thinking about allochthonous/autochthonous from the floodplain perspective. Thanks for coming back to this in the discussion (14,5) as well!

*Thank you, we are glad to hear that this choice of terminology and framing worked well for this manuscript.*

4,1 – Access roads and trails here as well, no? I know the winds can be pretty remote, but it's nice to get a feeling for direct human impacts as mentioned for logging in the MF Snoqualmie basin (even if they are minimal).

*Thanks for catching this omission. The Big Sandy has been grazed in the lower reaches (although we didn't notice any particularly incised streams or rapidly eroding banks that might indicate intensive over-grazing) and has an access road crossing the basin. We have noted this in the field site description.*

9,3 – I really appreciated these summaries at the end of each paragraph. Great way to stay in results mode but not loose sight of overall trends.

*This is great to hear. We are glad these summary paragraphs were effective.*

10,6 – Respiration is a mechanism. Consider restating. Perhaps "stabilization and loss mechanisms"?

*Good point. We have revised according your suggestion. This also fits better with our revisions to the conceptual model (Figure 3).*

10,9 – Missing a space: "2018),which"

*We have added a space.*

13,10 – As written, could be read as inputs are regulated by storage. Could restate as "the fate of OC inputs are regulated. . ."

*Reviewer 1 also commented on this sentence. We have revised based on both your comments to clarify that we mean that OC inputs are regulated by storage and OC processing in soil to determine OC concentrations in floodplain soil.*

19, Figure 1 – coordinates?

*We have added coordinates for the approximate centers of each basin to the figure caption.*

21, Figure 3 – see major comment #2 above.

*We have revised Figure 3 according to your and Reviewer 1's suggestions. Please see our response to major comment #2 above.*

[revised manuscript text omitted]

---

## Editor Comment (EC1) · R. G. Hilton (Editor) · 28 Sep 2018

Associate Editor comments following peer review of "Geomorphic regulation of floodplain soil organic carbon concentration in watersheds of the Rocky and Cascade Mountains, USA", by Scott & Wohl

I have now had the opportunity to read your manuscript in detail, before examining the comments made by the two referees. I concur with their reports, there are novel and interesting elements to this research and the paper is clearly suitable for ESurf.

I agree with all of the reviewers' recommendations. The replies you have posted suggest a revised version can address these comments. Please take this opportunity again to reflect on revisions (the conceptual diagram is still somewhat hard to link to the study).

My own reading of the manuscript has raised an important additional point. Organic carbon (%OC) concentration was quantified using measurements of loss on ignition (LOI). While this is still relatively common in some fields, it does have known (and potentially large, up to several 10s of %) biases for measurement of %OC. If the study had quantified organic carbon stocks (e.g. gC m-2) and/or fluxes (gC / yr) this would have been an issue: the results would not be comparable to methods which analyse %OC using combustion and elemental analyser methods. Here, the study focuses on a comparison of %OC between two study sites. Therefore, broadly speaking the %LOI approach should allow patterns and differences to be delineated. However, I would encourage future work that employs a more direct quantification of %OC in sediments and soils.

With this in mind, the manuscript needs more details on the resultant uncertainties that derive from using the %LOI method to get %OC. These are:

1) Structural clay water content is accounted for using a published approach. Can you add a sentence or two to explain the size of this correction (i.e. was the average proportion of the LOI weight loss attributed to this factor)?

2) Because this study focuses on site comparison, can you please specify how much this clay water correction factor varied between samples, and between locations. It is important to establish whether the LOI method introduces biases into the %OC estimates and subsequent analysis.

3) There are 10 measurements of %OC from CHN analysis (following carbonate removal). It would be useful to explain how these measurements differed from the %LOI approach on the same samples, and feed this into the analysis of uncertainty derived from the LOI method.

Other comments:

Pg1, L8 – I think you need to add "in floodplain soils" or similar to the end of the first sentence.

Pg1, L15 – 'differences' instead of 'trends'

Pg2, L31 – "...concentration of soil"

Pg3, L3 – its coming later, but it would be useful to briefly explain how these two locations differ. Indeed, a summary table would be a useful way to contrast the main sites shown in Figure 2.

Pg3, L18 – For the review article citation, please specify here that all relevant details are provided in this manuscript (and make sure they are). Or if the paper is on a pre-print server those details can be provided.

Pg6 – I appreciate the careful discussion of the LOI method for calculating %OC. However, there needs to be a little bit more detail on the uncertainties associated with the LOI method. 1) Can you add some more information on the clay content corrections (outline the degree of the correction, and assess whether they are systematic in any way between different locations, or across environmental gradients). 2) Please provide an estimate of uncertainty (precision and accuracy) for the %OC derived from LOI. 3) use the CHN %OC analyses to assess the accuracy of the LOI proxy for the small sample set where you have both measurements (n = 10).

Pg7, L2 – 'likely still be accurate' – this phrase can be qualified using the measurements of %OC from CHN versus %OC from LOI. Please do so (see comment above).

Pg12, L21 – it would be useful somewhere here to summarise some of the available geochemical approaches which may be used to examine in more detail the provenance and processing of organic matter (e.g. stable isotopes, radiocarbon activity, biomarkers, isotope composition of plant wax biomarkers) in the floodplain sediments.

Figure 1 – shows hillshade not topography, please edit the caption

**ESurfD**

---

## Author Comment (AC3) · 4 Oct 2018

**Response to Associate Editor**

We are grateful for the constructive and detailed commentary provided by the associate editor, and for providing us the opportunity to revise the work for publication. We have revised the manuscript to address the associate editor comments and responded to those comments below in *italics*. Our revised manuscript with changes tracked is shown below this response.

Associate Editor Comments
Associate Editor comments following peer review of "Geomorphic regulation of floodplain soil organic carbon concentration in watersheds of the Rocky and Cascade Mountains, USA", by Scott & Wohl

I have now had the opportunity to read your manuscript in detail, before examining the comments made by the two referees. I concur with their reports, there are novel and interesting elements to this research and the paper is clearly suitable for ESurf.

I agree with all of the reviewers' recommendations. The replies you have posted suggest a revised version can address these comments. Please take this opportunity again to reflect on revisions (the conceptual diagram is still somewhat hard to link to the study).

*We feel that the conceptual diagram, as it has been revised, strikes a balance between sticking to the results of our study and attempting to generalize those results. While we could make it more strictly related to this study, we feel that its current design is appropriate for our objective of trying to provide a conceptualization of floodplain soil OC dynamics. However, we have added a clause to the caption of this figure noting more explicitly that it is based on both this study and other literature.*

My own reading of the manuscript has raised an important additional point. Organic carbon (%OC) concentration was quantified using measurements of loss on ignition (LOI). While this is still relatively common in some fields, it does have known (and potentially large, up to several 10s of %) biases for measurement of %OC. If the study had quantified organic carbon stocks (e.g. gC m-2) and/or fluxes (gC / yr) this would have been an issue: the results would not be comparable to methods which analyse %OC using combustion and elemental analyser methods. Here, the study focuses on a comparison of %OC between two study sites. Therefore, broadly speaking the %LOI approach should allow patterns and differences to be delineated. However, I would encourage future work that employs a more direct quantification of %OC in sediments and soils.

*We appreciate your suggestion, but respectfully disagree with your comment that LOI has known biases for %OC measurement, based on our review of the literature comparing LOI to elemental analysis methods. While LOI does estimate OM, and hence requires a correction factor to estimate OC, we are unaware of any evidence that corrected LOI estimates (to obtain OC) are biased when compared to elemental analysis. Tests comparing LOI and elemental analysis all point to potentially lower precision of LOI estimates (e.g., Abella and Zimmer, 2007), but none*

*point to any systematic biases due to LOI when an OM to OC correction factor is applied* (Abella and Zimmer, 2007; Sleutel et al., 2007)*, especially when structural water content is appropriately corrected for* (Hoogsteen et al., 2015; Wang et al., 2011) *and carbonate concentrations are low, as they were in our study sites* (Wright et al., 2008)*. All of these studies conclude that LOI-based estimations of OC content are indeed comparable to those obtained by a CHN analyzer, albeit with some sacrifice in variability. We welcome suggestions of literature that show a bias (as opposed to simply higher variability) in LOI, but in lieu of that, we stand by our methodological decision. We chose LOI to balance cost (we had to analyze 293 soil samples for this study) and estimation accuracy. Our choice is further justified by the fact that we were able to robustly determine the controls on OC content in floodplain soils with statistical analyses despite the potentially higher variability in our OC estimates introduced by LOI.*

With this in mind, the manuscript needs more details on the resultant uncertainties that derive from using the %LOI method to get %OC. These are:

1) Structural clay water content is accounted for using a published approach. Can you add a sentence or two to explain the size of this correction (i.e. was the average proportion of the LOI weight loss attributed to this factor)?

*We have added a sentence to the end of paragraph 2 in section 2.4 that details this information. We decided to calculate the clay water correction magnitude as a proportion of the sample OC estimate (with the correction). While this is comparable to the proportion of the LOI weight loss, this perhaps more explicitly shows the magnitude of the correction factor with regard to the final OC estimate. We have reported this value as a range and a 95% confidence interval on the median for each basin studied.*

2) Because this study focuses on site comparison, can you please specify how much this clay water correction factor varied between samples, and between locations. It is important to establish whether the LOI method introduces biases into the %OC estimates and subsequent analysis.

*Please see our response to the previous comment. We have provided values for each basin separately, allowing for comparison. The 95% confidence intervals on the median estimates for the magnitude of the clay water content correction as a proportion of total OC content for each basin overlap substantially, indicating that this correction was similar between basins. Please also see our response to your comment above regarding LOI biases regarding that issue.*

3) There are 10 measurements of %OC from CHN analysis (following carbonate removal). It would be useful to explain how these measurements differed from the %LOI approach on the same samples, and feed this into the analysis of uncertainty derived from the LOI method.

*The CHN measurements were not conducted to represent a robust test of LOI as a method of estimating OC content, as we explicitly analyzed only samples that could potentially be calcium-carbonate-rich. However, we agree that it would be useful to compare these CHN measurements to our LOI estimates of OC. As such, we have added a sentence detailing the 95% confidence interval on the median difference between the LOI and CHN estimate for these 10 samples. This*

*95% confidence interval includes 0, which suggests a lack of bias in our LOI estimates. We did not conduct repeated analyses with any single method of any samples, and hence are unable to evaluate the variance in our LOI estimates. We have also added the CHN %OC measurements to our supplementary data table.*

Other comments:

Pg1, L8 – I think you need to add "in floodplain soils" or similar to the end of the first sentence.

*Good point, we have made the suggested change.*

Pg1, L15 – 'differences' instead of 'trends'

*We have made the suggested change.*

Pg2, L31 – ": : :concentration of soil"

*We have made the suggested addition here and elsewhere in the manuscript to improve clarity.*

Pg3, L3 – its coming later, but it would be useful to briefly explain how these two locations differ. Indeed, a summary table would be a useful way to contrast the main sites shown in Figure 2.

*We have added to this sentence to explain that the basins differ primarily in terms of hydroclimatic regime and vegetation characteristics. Instead of a summary table, we now show key basin characteristics in Figure 1 (showing field sites), to make it simpler for readers to quickly compare the two basins.*

Pg3, L18 – For the review article citation, please specify here that all relevant details are provided in this manuscript (and make sure they are). Or if the paper is on a pre-print server those details can be provided.

*This article was published, and the citation has been changed accordingly. We feel that all relevant details are indeed provided in this manuscript. In fact, this citation is more to explain why the methods sections of these two papers are so similar (in case readers happen to read both), instead of trying to save space in this paper by directing readers to the paper cited here.*

Pg6 – I appreciate the careful discussion of the LOI method for calculating %OC. However, there needs to be a little bit more detail on the uncertainties associated with the LOI method. 1) Can you add some more information on the clay content corrections (outline the degree of the correction, and assess whether they are systematic in any way between different locations, or across environmental gradients). 2) Please provide an estimate of uncertainty (precision and accuracy) for the %OC derived from LOI. 3) use the CHN %OC analyses to assess the accuracy of the LOI proxy for the small sample set where you have both measurements (n = 10).

*Please see our responses to your major comment above. We have added information regarding clay held water content corrections (their magnitude and whether they vary between basins). We cannot estimate the precision of our LOI-derived %OC measurements, as we did not perform replicate analyses. We can assess the accuracy, comparing LOI %OC estimates to CHN %OC estimates, as you suggest in point 3. We have done this, and added language explaining it to the end of section 2.4.*

Pg7, L2 – 'likely still be accurate' – this phrase can be qualified using the measurements of %OC from CHN versus %OC from LOI. Please do so (see comment above).

*Please see our responses to your above comments. We have done so, and we detail our changes above.*

Pg12, L21 – it would be useful somewhere here to summarise some of the available geochemical approaches which may be used to examine in more detail the provenance and processing of organic matter (e.g. stable isotopes, radiocarbon activity, biomarkers, isotope composition of plant wax biomarkers) in the floodplain sediments.

*We feel that such a summary would be out of the scope of this paper. In addition, we are not experts in using geochemical approaches to accomplish such a goal, so we feel that we likely would not do such a summary justice in suggesting valid methods. As such, we have chosen to make no changes with regard to this comment.*

Figure 1 – shows hillshade not topography, please edit the caption

*We have edited the caption accordingly.*

References Cited

Abella, S.R., Zimmer, B.W., 2007. Estimating Organic Carbon from Loss-On-Ignition in Northern Arizona Forest Soils. Soil Sci. Soc. Am. J. 71, 545. https://doi.org/10.2136/sssaj2006.0136

Hoogsteen, M.J.J., Lantinga, E.A., Bakker, E.J., Groot, J.C.J., Tittonell, P.A., 2015. Estimating soil organic carbon through loss on ignition: Effects of ignition conditions and structural water loss. Eur. J. Soil Sci. 66, 320–328. https://doi.org/10.1111/ejss.12224

Sleutel, S., De Neve, S., Singier, B., Hofman, G., 2007. Quantification of organic carbon in soils: A comparison of methodologies and assessment of the carbon content of organic matter. Commun. Soil Sci. Plant Anal. 38, 2647–2657. https://doi.org/10.1080/00103620701662877

Wang, Q., Li, Y., Wang, Y., 2011. Optimizing the weight loss-on-ignition methodology to quantify organic and carbonate carbon of sediments from diverse sources. Environ. Monit. Assess. 174, 241–257. https://doi.org/10.1007/s10661-010-1454-z

Wright, A.L., Wang, Y., Reddy, K.R., 2008. Loss-on-Ignition Method to Assess Soil Organic Carbon in Calcareous Everglades Wetlands. Commun. Soil Sci. Plant Anal. https://doi.org/10.1080/00103620802432931

[revised manuscript text omitted]